# Recurrent Registration Neural Networks for Deformable Image Registration

**Robin Sandkühler**
Department of Biomedical Engineering
University of Basel, Switzerland
robin.sandkuehler@unibas.ch

**Simon Andermatt**
Department of Biomedical Engineering
University of Basel, Switzerland
simon.andermatt@unibas.ch

**Grzegorz Bauman**
Division of Radiological Physics
Department of Radiology
University of Basel Hospital, Switzerland
grzegorz.bauman@usb.ch

**Sylvia Nyilas**
Pediatric Respiratory Medicine
Department of Pediatrics
Inselspital, Bern University Hospital
University of Bern, Switzerland
sylvia.nyilas@insel.ch

**Christoph Jud**
Department of Biomedical Engineering
University of Basel, Switzerland
christoph.jud@unibas.ch

**Philippe C. Cattin**
Department of Biomedical Engineering
University of Basel, Switzerland
philippe.cattin@unibas.ch

## Abstract

Parametric spatial transformation models have been successfully applied to image registration tasks. In such models, the transformation of interest is parameterized by a fixed set of basis functions as for example B-splines. Each basis function is located on a fixed regular grid position among the image domain because the transformation of interest is not known in advance. As a consequence, not all basis functions will necessarily contribute to the final transformation which results in a non-compact representation of the transformation. We reformulate the pairwise registration problem as a recursive sequence of successive alignments. For each element in the sequence, a local deformation defined by its position, shape, and weight is computed by our recurrent registration neural network. The sum of all local deformations yield the final spatial alignment of both images. Formulating the registration problem in this way allows the network to detect non-aligned regions in the images and to learn how to locally refine the registration properly. In contrast to current non-sequence-based registration methods, our approach iteratively applies local spatial deformations to the images until the desired registration accuracy is achieved. We trained our network on 2D magnetic resonance images of the lung and compared our method to a standard parametric B-spline registration. The experiments show, that our method performs on par for the accuracy but yields a more compact representation of the transformation. Furthermore, we achieve a speedup of around 15 compared to the B-spline registration.

## 1 Introduction

Image registration is essential for medical image analysis methods, where corresponding anatomical structures in two or more images need to be spatially aligned. The misalignment often occurs in images from the same structure between different imaging modalities (CT, SPECT, MRI) or during

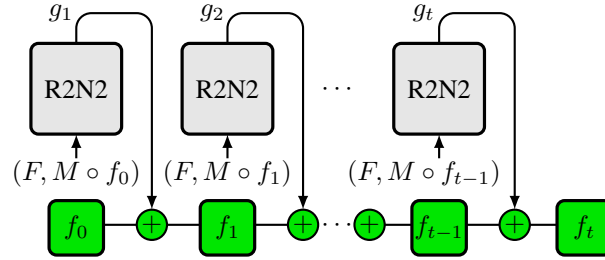

Figure 1: Sequence-based registration process for pairwise deformable image registration of a fixed image $F$ and a moving image $M$.

the acquisition of dynamic time series (2D+t, 4D). An overview of registration methods and their different categories is given in [24]. In this work, we will focus on parametric transformation models in combination with learning-based registration methods. There are mainly two major classes of parametric transformation models used in medical image registration. The first class are the dense transformation models or so-called optical-flow [11]. Here, the transformation of each pixel in the image is directly estimated (Figure 2a). The second class of models are interpolating transformation models (Figure 2b). Interpolating transformation models approximate the transformation between both images with a set of fixed basis functions (e.g. Gaussian, B-spline) among a fixed grid of the image domain [22, 27, 15, 14]. These models reduce the number of free parameters for the optimization, but restrict the space of admissible transformations. Both transformation models have advantages and disadvantages. Dense models allow preservation of local discontinuities of the transformation, while the interpolating models achieve a global smoothness if the chosen basis function is smooth.

Although the computation time for the registration has been reduced in the past, image registration is still computationally costly, because a non-linear optimization problem needs to be solved for each pair of images. In order to reduce the computation time and to increase the accuracy of the registration result, learning-based registration methods have been recently introduced. As the registration is now separated in a training and an inference part, a major advantage in computation time for the registration is achieved. A detailed overview of deep learning methods for image registration is given in [8]. The FlowNet [6] uses a convolutional neural network (CNN) to learn the optical flow between two input images. They trained their network in a supervised fashion using ground-truth transformations from synthetic data sets. Based on the idea of the spatial transformer networks [13], unsupervised learning-based registration methods were introduced [5, 4, 26, 12]. All of these methods have in common that the output of the network is directly the final transformation. In contrast, sequence-based methods do not estimate the final transformation in one step but rather in a series of transformations based on observations of the previous transformation result. This process is iteratively continued until the desired accuracy is achieved. Applying a sequence of local or global deformations is inspired by how a human would align two images by applying a sequence of local or global deformations. Sequence-based methods for rigid [18, 20] and for deformable [17] registration using reinforcement learning methods were introduced in the past. However, the action space for deformable image registration can be very large and the training of deep reinforcement learning methods is still very challenging.

In this work, we present the *Recurrent Registration Neural Network* (R2N2), a novel sequence-based registration method for deformable image registration. Figure 1 shows the registration process with the R2N2. Instead of learning the transformation as a whole, we iteratively apply a network to detect local differences between two images and determine how to align them using a parameterized local deformation. Modeling the final transformation of interest as a sequence of local parametric transformations instead of a fixed set of basis functions enables our method to extend the space of admissible transformations, and to achieve a global smoothness. Furthermore, we are able to achieve a compact representation of the final transformation. As we define the resulting transformation as a recursive sequence of local transformations, we base our architecture on recurrent neural networks. To the best of our knowledge, recurrent neural networks are not used before for deformable image registration.

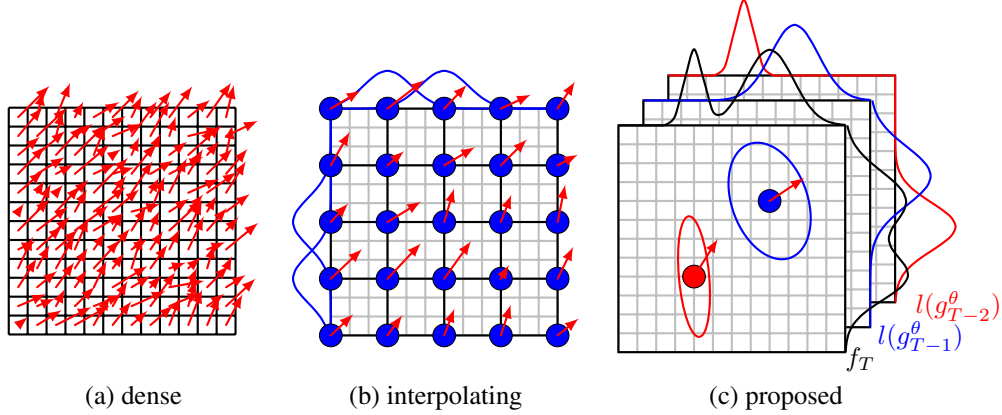

|  (a) dense | (b) interpolating | (c) proposed |

Figure 2: Dense, interpolating, and proposed transformation models.

## 2  Background

Given two images that need to be aligned, the fixed image $F : \mathcal{X} \to \mathbb{R}$ and the moving image $M : \mathcal{X} \to \mathbb{R}$ on the image domain $\mathcal{X} \subset \mathbb{R}^d$, the pairwise registration problem can be defined as a regularized minimization problem

$$f^* = \arg\min_{f} \mathcal{S}[F, M \circ f] + \lambda \mathcal{R}[f]. \tag{1}$$

Here, $f^* : \mathcal{X} \to \mathbb{R}^d$ is the transformation of interest and a minimizer of (1). The image loss $\mathcal{S} : \mathcal{X} \times \mathcal{X} \to \mathbb{R}$ determines the image similarity of $F$ and $M \circ f$, with $(M \circ f)(x) = M(x + f(x))$. In order to restrict the transformation space by using prior knowledge of the transformation, a regularization loss $\mathcal{R} : \mathbb{R}^d \to \mathbb{R}$ and the regularization weight $\lambda$ are added to the optimization problem. The regularizer is chosen depending on the expected transformation characteristics (e.g. global smoothness or piece-wise smoothness).

### 2.1  Transformation

In order to optimize (1) a transformation model $f_\theta$ is needed. The minimization problem then becomes

$$\theta^* = \arg\min_{\theta} \mathcal{S}[F, M \circ f_\theta] + \lambda \mathcal{R}[f_\theta], \tag{2}$$

where $\theta$ are the parameters of the transformation model. There are two major classes of transformation models used in image registration: dense and interpolating. In the dense case, the transformation at position $x$ in the image is defined by a displacement vector

$$f_\theta(x) = \theta_x, \tag{3}$$

with $\theta_x = (\vartheta_1, \vartheta_2, \ldots, \vartheta_d) \in \mathbb{R}^d$. For the interpolating case the transformation at position $x$ is normally defined in a smooth basis

$$f_\theta(x) = \sum_{i}^{N} \theta_i k(x, c_i). \tag{4}$$

Here, $\{c_i\}_{i=1}^{N}, c_i \in \mathcal{X}$ are the positions of the fixed regular grid points in the image domain, $k : \mathcal{X} \times \mathcal{X} \to \mathbb{R}$ the basis function, and $N$ the number of grid points. The transformation between the control points $c_i$ is an interpolation of the control point values $\theta_i \in \mathbb{R}^d$ with the basis function $k$. A visualization of a dense and an interpolating transformation model is shown in Figure 2.

### 2.2  Recurrent Neural Networks

Recurrent Neural Networks (RNNs) are a class of neural networks designed for sequential data. A simple RNN has the form

$$h_t = \phi(W x_t + U h_{t-1}), \tag{5}$$

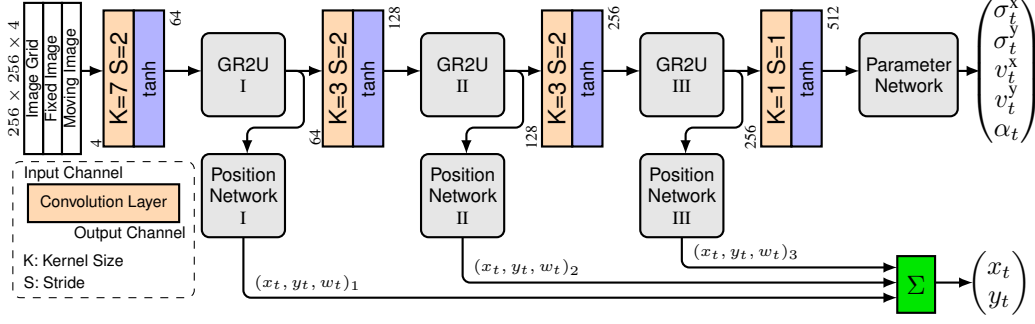

Figure 3: Network architecture of the presented Recurrent Registration Neural Network.

where $W$ is a weighting matrix of the input at time $t$, $U$ is the weight matrix of the last output at time $t-1$, and $\phi$ is an activation function like the hyperbolic tangent or the logistic function. Since the output at time $t$ directly depends on the weighted previous output $h_{t-1}$, RNNs are well suited for the detection of sequential information which is encoded in the sequence itself. RNNs provide an elegant way of incorporating the whole previous sequence without adding a large number of parameters. Besides the advantage of RNNs for sequential data, there are some difficulties to address e.g. the problem to learn long-term dependencies. The long short-term memory (LSTM) architecture was introduced in order to overcome these problems of the basic RNN [10]. A variation of the LSTM, the gated recurrent unit (GRU) was presented by [3].

## 3 Methods

In the following, we will present our *Recurrent Registration Neural Network* (R2N2) for the application of sequence-based pairwise medical image registration of 2D images.

### 3.1 Sequence-Based Image Registration

Sequence-based registration methods do not estimate the final transformation in one step but rather in a series of local transformations. The minimization problem for the sequence-based registration is given as

$$\theta^* = \arg\min_{\theta} \frac{1}{T} \sum_{t=1}^{T} \mathcal{S}[F, M \circ f_t^{\theta}] + \lambda \mathcal{R}[f_T]. \tag{6}$$

Compared to the registration problem (2) the transformation $f_t^{\theta}$ is now defined as a recursive function of the form

$$f_t^{\theta}(x, F, M) = \begin{cases} 0, & \text{if } t = 0, \\ f_{t-1}^{\theta} + l(x, g_{\theta}(F, M \circ f_{t-1}^{\theta})) & \text{else.} \end{cases} \tag{7}$$

Here, $g_{\theta}$ is the function that outputs the parameter of the next local transformation given the two images $F$ and $M \circ f_t^{\theta}$. In each time step $t$, a local transformation $l : \mathcal{X} \times \mathcal{X} \to \mathbb{R}^2$ is computed and added to the transformation $f_t^{\theta}$. After transforming the moving image $M$ with $f_t^{\theta}$, the result is used as input for the next time step, in order to compute the next local transformation as shown in Figure 1. This procedure is repeated until both input images are aligned. We define a local transformation as a Gaussian function

$$l(x, \tilde{x}_t, \Gamma_t, v_t) = v_t \exp\left(-\frac{1}{2}(x - \tilde{x}_t)^T \Sigma(\Gamma_t)^{-1}(x - \tilde{x}_t)\right), \tag{8}$$

where $\tilde{x}_t = (x_t, y_t) \in \mathcal{X}$ is the position, $v_t = (v_t^{\mathrm{x}}, v_t^{\mathrm{y}}) \in [-1, 1]^2$ the weight, and $\Gamma_t = \{\sigma_t^{\mathrm{x}}, \sigma_t^{\mathrm{y}}, \alpha_t\}$ the shape parameter with

$$\Sigma(\Gamma_t) = \begin{bmatrix} \cos(\alpha_t) & -\sin(\alpha_t) \\ \sin(\alpha_t) & \cos(\alpha_t) \end{bmatrix} \begin{bmatrix} \sigma_t^{\mathrm{x}} & 0 \\ 0 & \sigma_t^{\mathrm{y}} \end{bmatrix} \begin{bmatrix} \cos(\alpha_t) & -\sin(\alpha_t) \\ \sin(\alpha_t) & \cos(\alpha_t) \end{bmatrix}^{\mathrm{T}}. \tag{9}$$

Here, $\sigma_t^x, \sigma_t^y \in \mathbb{R}_{>0}$ control the width and $\alpha_t \in [0, \pi]$ the rotation of the Gaussian function. The output of $g_\theta$ is defined as $g_\theta = \{\tilde{x}_t, \Gamma_t, v_t\}$. Compared to the interpolating registration model shown in Figure 2b, the position $\tilde{x}_t$ and shape $\Gamma_t$ of the basis functions are not fixed during the registration in our method (Figure 2c).

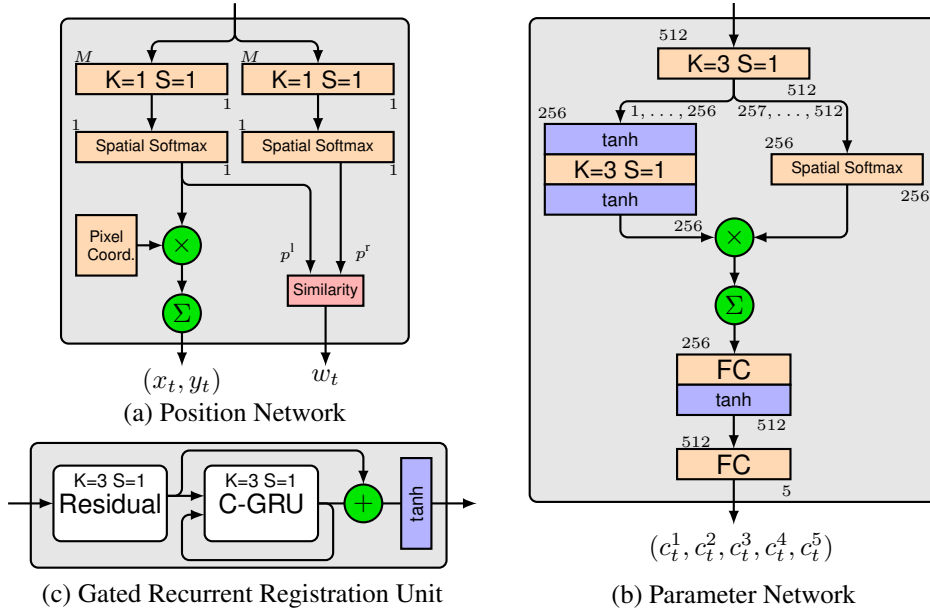

Figure 4: Architectures for the position network, the parameter network, and the gated recurrent registration unit.

## 3.2 Network Architecture

We developed a network architecture to approximate the unknown function $g_\theta$, where $\theta$ are the parameters of the network. Since the transformation of the registration is defined as a recursive sequence, we base our network up on GRUs due to their efficient gated architecture. An overview of the complete network architecture is shown in Figure 3. The input of the network are two images, the fixed image $F$ and the moving image $M \circ f_t$. As suggested in [19], we attached the position of each pixel as two additional coordinate channels to improve the convolution layers for the handling of spatial representations. Our network contains three major sub-networks to generate the parameters of the local transformation: the gated recurrent registration unit (GR2U), the position network, and the parameter network.

**Gated Recurrent Registration Unit**   Our network contains three GR2U for different spatial resolutions ($128 \times 128$, $64 \times 64$, $32 \times 32$). Each GR2U has an internal structure as shown in Figure 4c. The input of the GR2U block is passed through a residual network, with three stacked residual blocks [9]. If not stated otherwise, we use the hyperbolic tangent as activation function in the network. The core of each GR2U is the C-GRU block. For this, we adopt the original GRU equations shown in [3] in order to use convolutions instead of a fully connected layer as presented in [1]. In contrast to [1], we adapt the proposal gate (12) for use with convolutions, but without factoring $r_j$ out of the convolution. The C-GRU is then defined by:

$$r^j = \psi\left(\sum_i^I \left(x * w_r^{i,j}\right) + \sum_k^J \left(h_{t-1}^k * u_r^{k,j}\right) + b_r^j\right), \tag{10}$$

$$z^j = \psi\left(\sum_i^I \left(x * w_z^{i,j}\right) + \sum_k^J \left(h_{t-1}^k * u_z^{k,j}\right) + b_z^j\right), \tag{11}$$

$$\tilde{h}_t^j = \phi\left(\sum_i^I \left(x * w^{i,j}\right) + \sum_k^J \left((r_j \odot h_{t-1}^k) * u^{k,j}\right) + b^j\right), \tag{12}$$

$$h_t^j = (1 - z^j) \odot h_{t-1}^j + z_j \odot \tilde{h}_t^j. \tag{13}$$

Here, $r$ represents the reset gate, $z$ the update gate, $\tilde{h}_t$ the proposal state, and $h_t$ the output at time $t$. We define $\phi(\cdot)$ as the hyperbolic tangent, $\psi(\cdot)$ represents the logistic function, and $\odot$ is the Hadamard product. The convolution is denoted as $*$ and $u_., w_., b_.$ are the parameters to be learned. The indices $i, j, k$ correspond to the input and output/state channel index. We also applied a skip connection from the output of the residual block to the output of the C-GRU.

**Position Network**  The architecture of the position network is shown in Figure 4a and contains two paths. In the left path, the position of the local transformation $x_t^n$ is calculated using a convolution layer followed by the *spatial softmax* function [7]. Here, $n$ is the level of the spatial resolution. The spatial softmax function is defined as

$$p_k(c_{ij}^k) = \frac{\exp(c_{ij}^k)}{\sum_{i'} \sum_{j'} \exp(c_{i'j'}^k)}, \tag{14}$$

where $i$ and $j$ are the spatial indices of the $k$-th feature map $c$. The position is then calculated by

$$x_t^n = \left(\sum_i \sum_j p(c_{ij}) X_{ij}^n, \sum_i \sum_j p(c_{ij}) Y_{ij}^n\right), \tag{15}$$

where $(X_{ij}^n, Y_{ij}^n) \in \mathcal{X}$ are the coordinates of the image pixel grid. As shown in Figure 3 an estimate of the current transformation position is computed on all three spatial levels. The final position is calculated as a weighted sum

$$\tilde{x}_t = \frac{\sum_n^3 x_t^n w_t^n}{\sum_n^3 w_t^n}. \tag{16}$$

The weights $w_t^n \in \mathbb{R}$ are calculated on the right side of the position block. For this, a second convolution layer and a second *spatial softmax* layer are applied to the input of the block. We calculated the similarity of the left spatial softmax $p^l(c_{ij})$ and the right spatial softmax $p^r(c_{ij})$ as the weight of the position at each spatial location

$$w_t^n = 2 - \sum_i \sum_j \left|p^l(c_{ij}) - p^r(c_{ij})\right|. \tag{17}$$

This weighting factor can be interpreted as certainty measure of the estimation of the current position at each spatial resolution.

**Parameter Network**  The parameter network is located at the end of the network. Its detailed structure is shown in Figure 4b. The input of the parameter block is first passed through a convolution layer. After the convolution layer, the first half of the output feature maps is passed through a second convolution layer. The second half is applied to a *spatial softmax* layer. For each element in both outputs, a point-wise multiplication is applied, followed by an average pooling layer down to a spatial resolution of $1 \times 1$. We use a fully connected layer with one hidden layer in order to reduce the output to the number of needed parameters. The final output parameters are then defined as

$$\sigma_t^x = \psi(c_t^1)\sigma_{max}, \quad \sigma_t^y = \psi(c_t^2)\sigma_{max}, \quad v_t^x = \phi(c_t^3), \quad v_t^y = \phi(c_t^4), \quad \alpha_t = \psi(c_t^5)\pi, \tag{18}$$

where $\phi(\cdot)$ is the hyperbolic tangent, $\psi(\cdot)$ the logistic function, and $\sigma_{max}$ the maximum extension of the shape.

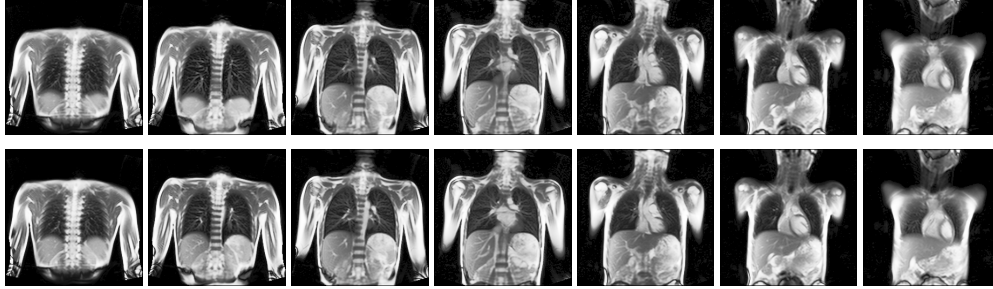

Figure 5: Maximum inspiration (top row) and maximum expiration (bottom row) for different slice positions of one patient from back to front.

## 4 Experiments and Results

**Image Data**  We trained our network on images of a 2D+t magnetic resonance (MR) image series of the lung. Due to the low proton density of the lung parenchyma in comparison to other body tissues as well as strong magnetic susceptibility effects, it is very challenging to acquire MR images with a sufficient signal-to-noise ratio. Recently, a novel MR pulse sequence called ultra-fast steady-state free precession (ufSSFP) was proposed [2]. ufSSFP allows detecting physiological signal changes in lung parenchyma caused by respiratory and cardiac cycles, without the need for intravenous contrast agents or hyperpolarized gas tracers. Multi-slice 2D+t ufSSFP acquisitions are performed in free-breathing.

For a complete chest volume coverage, the lung is scanned at different slice positions as shown in Figure 5. At each slice position, a dynamic 2D+t image series with 140 images is acquired. For the further analysis of the image data, all images of one slice position need to be spatially aligned. We choose the image which is closest to the mean respiratory cycle as fixed image of the series. The other images of the series are then registered to this image. Our data set consists of 48 lung acquisitions of 42 different patients. Each lung scan contains between 7 and 14 slices. We used the data of 34 patients for the training set, 4 for the evaluation set, and 4 for the test set.

**Network Training**  The network was trained in an unsupervised fashion for $\sim 180{,}000$ iterations with a fixed sequence length of $t = 25$. Figure 6 shows an overview of the training procedure. We used the Adam optimizer [16] with the AMSGrad option [21] and a learning rate of $0.0001$. The maximum shape size is set to $\sigma_{\max} = 0.3$ and the regularization weight to $\lambda_{\text{R2N2}} = 0.1$. For the regularization of the network parameter, we use a combination of [25] particularly the use of Gaussian multiplicative noise and dropconnect [28]. We apply multiplicative Gaussian noise $\mathcal{N}(1, \sqrt{0.5}/0.5)$ to the parameter of the proposal and the output of the C-GRU. As image loss function $\mathcal{S}$ the mean squared error (MSE) loss is used and as transformation regularizer $\mathcal{R}$ the isotropic total variation (TV). The training of the network was performed on an NVIDIA Tesla V100 GPU.

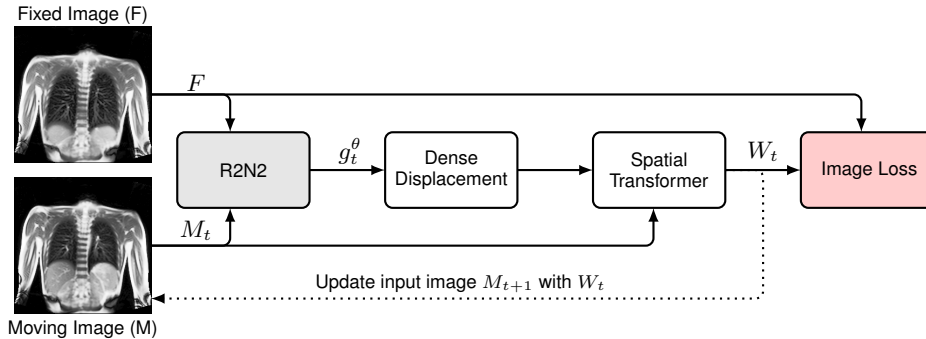

Figure 6: Unsupervised training setup ($W_t$ is the transformed moving image).

Table 1: Mean target registration error (TRE) for the proposed method R2N2 and a standard B-spline registration (BS) for the test data set in millimeter. The small number is the maximum TRE for all images for this slice.

| Patient | Slice 1 | Slice 2 | Slice 3 | Slice 4 | Slice 5 | Slice 6 | Slice 7 | Slice 8 | mean |
|---|---|---|---|---|---|---|---|---|---|
| R2N2 | 1.26 1.85 | 1.08 2.14 | 1.13 1.82 | 1.23 2.58 | 1.47 2.74 | 1.12 1.51 | 0.92 1.33 | 1.04 1.87 | 1.16 |
| BS | 1.28 1.81 | 1.16 2.0 | 1.40 2.52 | 1.15 2.67 | 0.96 1.71 | 0.99 1.41 | 0.84 1.14 | 1.02 1.65 | 1.10 |
| R2N2 | 0.84 1.99 | 0.92 2.49 | 0.79 1.04 | 0.81 1.2 | 0.74 1.43 | – | – | – | 0.82 |
| BS | 1.50 5.07 | 0.69 1.73 | 0.73 1.05 | 0.77 1.13 | 0.86 1.76 | – | – | – | 0.91 |
| R2N2 | 1.65 3.88 | 1.06 2.55 | 0.86 2.08 | 0.83 1.48 | 0.80 1.39 | 0.73 1.08 | – | – | 0.99 |
| BS | 1.15 2.73 | 0.81 1.42 | 0.75 1.64 | 0.79 1.14 | 0.72 0.94 | 0.83 1.95 | – | – | 0.84 |
| R2N2 | 1.30 3.03 | 0.77 0.98 | 0.79 2.07 | 1.09 1.92 | 0.84 1.12 | – | – | – | 0.96 |
| BS | 1.09 3.15 | 0.78 1.01 | 0.73 1.73 | 1.09 2.5 | 0.79 1.13 | – | – | – | 0.90 |

**Experiments**   We compare our method against a standard B-spline registration method (BS) implemented in the AirLab framework [23]. The B-spline registration use three spatial resolutions (64, 128, 256) with a kernel size of (7, 21, 57) pixels. As image loss the MSE and as regularizer the isotropic TV is used, with the regularization weight $\lambda_{BS} = 0.01$. We use the Adam optimizer [16] with the AMSGrad option [21], a learning rate of $0.001$, and we perform $250$ iterations per resolution level.

From the test set we select 21 images of each slice position, which corresponds to one breathing cycle. We then select corresponding landmarks in all 21 images in order to compute the registration accuracy. The target registration error (TRE) of the registration is defined as the mean root square error of the landmark distance after the registration. The results in Table 1 show that our presented method performed on par with the standard B-spline registration in terms of accuracy. Since the slice positions are manually selected for each patient, we are not able to provide the same amount of slices for each patient. Despite that the image data is different at each slice position, we see a good generalization ability of our network to perform an accurate registration independently of the slice position at which the images are acquired. Our method achieve a compact representation of

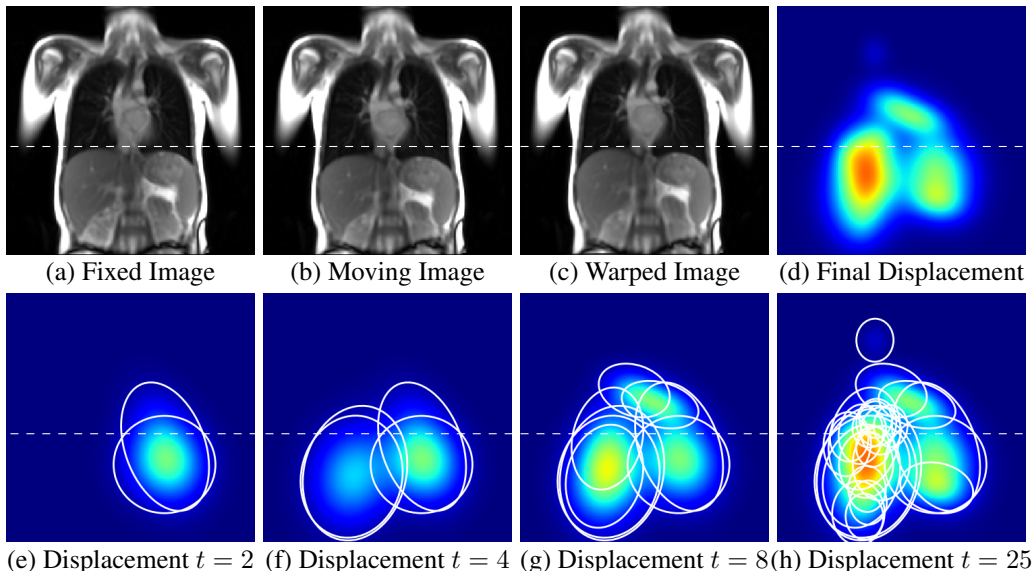

(a) Fixed Image     (b) Moving Image     (c) Warped Image     (d) Final Displacement

(e) Displacement $t = 2$  (f) Displacement $t = 4$  (g) Displacement $t = 8$ (h) Displacement $t = 25$

Figure 7: Top Row: Registration result of the proposed recurrent registration neural network for one image pair. Bottom Row: Sequence of local transformations after different time steps.

the final transformation, by using only $\sim 7.6\%$ of the amount of parameters than the final B-spline transformation. Here, the number of parameters of the network are not taken into account only

the number of parameters needed to describe the final transformation. For the evaluation of the computation time for the registration of one image pair, we run both methods on an NVIDIA GeForce GTX 1080. The computation of the B-spline registration takes $\sim 4.5$s compared to $\sim 0.3$s for our method.

An example registration result of our presented method is shown in Figure 7. It can be seen that the first local transformations the network creates are placed below the diaphragm (white dashed line) (Figure 7a), where the magnitude of the motion between the images is maximal. Also the shape and rotation of the local transformations are computed optimally in order to apply a transformation only at the liver and the lung and not on the rips. During the next time steps, we can observe that the shape of the local transformation is reduced to align finer details of the images (Figure 7g-h).

## 5   Conclusion

In this paper, we presented the *Recurrent Registration Neural Network* for the task of deformable image registration. We define the registration process of two images as a recursive sequence of local deformations. The sum of all local deformations yields the final spatial alignment of both images Our designed network can be trained end-to-end in an unsupervised fashion. The results show that our method is able to accurately register two images with a similar accuracy compared to a standard B-spline registration method. We achieve a speedup of $\sim 15$ for the computation time compared to the B-spline registration. In addition, we need only $\sim 7.6\%$ of the amount of parameters to describe the final transformation than the final transformation of the standard B-spline registration. In this paper, we have shown that our method is able to register two images in a recursive manner using a fixed number of steps. For future work we will including uncertainty measures for the registration result as a possible stopping criteria. This could then be used to automatically determine the number of steps needed for the registration. Furthermore, we will extend our method for the registration of 3D volumes.

## Acknowledgements

We would like to thank Oliver Bieri, Orso Pusterla (Division of Radiological Physics, Department of Radiology, University Hospital Basel, Switzerland), and Philipp Latzin (Pediatric Respiratory Medicine, Department of Pediatrics, Inselspital, Bern University Hospital, University of Bern, Switzerland) for there support during the development of this work. Furthermore, we thank the Swiss National Science Foundation for funding this project (SNF 320030_149576).

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
