[Supplementary Material · neurips_2019_supp.pdf]

# Recurrent Registration Neural Networks for Deformable Image Registration: Supplementary Material

**Robin Sandkühler**
Department of Biomedical Engineering
University of Basel, Switzerland
robin.sandkuehler@unibas.ch

**Simon Andermatt**
Department of Biomedical Engineering
University of Basel, Switzerland
simon.andermatt@unibas.ch

**Grzegorz Bauman**
Division of Radiological Physics
Department of Radiology
University of Basel Hospital, Switzerland
grzegorz.bauman@usb.ch

**Sylvia Nyilas**
Pediatric Respiratory Medicine
Department of Pediatrics
Inselspital, Bern University Hospital
University of Bern, Switzerland

**Christoph Jud**
Department of Biomedical Engineering
University of Basel, Switzerland
christoph.jud@unibas.ch

**Philippe C. Cattin**
Department of Biomedical Engineering
University of Basel, Switzerland
philippe.cattin@unibas.ch

## 1 Supplementary Material

In this document, we present additional information and results of the R2N2 registration algorithm.

# 2 Network Training

Algorithm 1 shows an overview of the training procedure of the presented *Recurrent Registration Neural Network* (R2N2).

---

**Algorithm 1** Training procedure of the R2N2

---

$\theta \leftarrow$ Initialize network parameters
$i \leftarrow 0$
**for** i < number of training iterations **do**
$\quad M_i, F_i \leftarrow$ Sample image pair from training set
$\quad f_0 \leftarrow 0 \qquad\quad$ Initialize transformation
$\quad \mathcal{L}_\mathcal{S} \leftarrow 0 \qquad\quad$ Initialize sequence image loss
$\quad W_0 \leftarrow M_i$
$\quad t \leftarrow 1$
$\quad$ **for** t < number of max transformations T **do**
$\quad\quad g_t \leftarrow \text{R2N2}(F_i, W_{t-1}, \theta) \quad$ Evaluate model
$\quad\quad f_t \leftarrow f_{t-1} + l(g_t) \qquad\quad$ Update transformation
$\quad\quad W_t \leftarrow M_i \circ f_t \qquad\quad$ Transform $M_i$
$\quad\quad \mathcal{L}_\mathcal{S} \leftarrow \mathcal{L}_\mathcal{S} + \mathcal{S}[F_i, W_t] \qquad$ Calculate image loss
$\quad\quad t \leftarrow t + 1$
$\quad$ **end for**
$\quad \mathcal{L}_\mathcal{R} \leftarrow \mathcal{R}[f_T] \qquad\qquad\qquad$ Calculate regularizer loss
$\quad \mathcal{L} \leftarrow \mathcal{L}_\mathcal{S} + \lambda \mathcal{L}_\mathcal{R} \qquad\qquad$ Final loss
$\quad$ Compute the gradient $\frac{\partial \mathcal{L}}{\partial \theta}$
$\quad$ Update the network parameters $\theta$
$\quad i \leftarrow i + 1$
**end for**

---

# 3 Results

In the following, we show detailed results of the presented *Recurrent Registration Neural Network* for the registration of one image pair. Figure 3-7 visualize the input and output of the R2N2 at each time step $t$ during the registration. We can observe a decreasing tendency of the shape parameter $\sigma_t$ (Figure 1) and the weight parameter $v_t$ (Figure 2) during the registration. That shows that our network achieves a coarse correction of large misalignments during the first steps and then continues with the finer ones. This registration procedure is very similar to how a human would align two images by applying a sequence of local deformations. The difference between the weight value $v_x$ and the weight value $v_x$ (Figure 2) can be explained by the fact that the major misalignment of the lung images are evoked by the breathing motion of the patient, which is oriented along the $y-$direction.

Figure 1: Shape size $\sigma_x$ and $\sigma_y$ of the local transformation outputted by the R2N2 at each time step $t$ for one image pair.

Figure 2: Wights $v_x$ and $v_y$ of the local transformation outputted by the R2N2 at each time step $t$ for one image pair.

Figure 3: Input and output for the R2N2 network for the time steps $0 - 4$ during the registration of one image pair. Here, $\circ$ is the transformation of the image on the left side with the transformation on the right side.

Figure 4: Input and output for the R2N2 network for the time steps $5-9$ during the registration of one image pair. Here, $\circ$ is the transformation of the image on the left side with the transformation on the right side.

Figure 5: Input and output for the R2N2 network for the time steps $10 - 14$ during the registration of one image pair. Here, ∘ is the transformation of the image on the left side with the transformation on the right side.

Figure 6: Input and output for the R2N2 network for the time steps $15 - 19$ during the registration of one image pair. Here, $\circ$ is the transformation of the image on the left side with the transformation on the right side.

Figure 7: Input and output for the R2N2 network for the time steps $20 - 24$ during the registration of one image pair. Here, $\circ$ is the transformation of the image on the left side with the transformation on the right side.