[Reviews · NeurIPS 2019]

Reviewer 1



This paper is well written, and presents a novel idea of pairwise image registration using RNNs that can be trained in an unsupervised manner. Registration performance is on par with the B-Spline Image registration method and is much faster than the latter. Authors have described their approach very well, and the provided convincing results as well as supplementary material that was helpful for the review.

Reviewer 2



1. The main advantage of this approach is its efficiency at inference time with comparable performance of B-spline based approach where an optimization is needed per registration. And it has, according to the authors, much less parameters to optimize. Please confirm if this understanding is correct? 2. What is the reason of making the choice of using multiple steps to gradually transform the moving image to the fixed one? Could the local transformation done in one step instead? For instance, the position network could directly predict K locations to transform in one step instead of prediction one location for K steps. What is the difference? 3. What is the reason of using Gaussian to perform the local transformation? Are there any other choices? 4. It seems odd to me that the parameter network is not explicitly made aware of the decision made by the position network as they would have to collaborate to perform the image transformation. If the GRU is needed to run very many steps, the positions predicted early may not be relevant to what is produced by the parameter network at the end. 5. The application of MRI chest is not properly motivated. Why it is important to solve such a task and how hard it is to solve it? 6. With the visualizations included, the registration tasks seem to be quite easy to solve. I would like to see if this model could be applied on more challenging tasks such a registering longitudinal MR/CT studies on the same patient from different timepoint. 7. Perhaps this is not very clear in the text, if I have a point in the moving image, say (x,y), how do I derive its corresponding point (x', y') in the fixed image with the trained model? 8. Can you describe in more details the principle of B-Spline method, as this is the only benchmark that you have compared against? ----------- Post author feedback: It is very clear to me that the main novelty is on achieving the speedup in inference while maintaining a good accuracy. The author's response resolves most of my concern around architecture choices and parameterization. There are still two issues that I didn't get satisfactory answers: 1. Why use multiple fixation instead of one? Why there's no experiments justifying such an important decision? All experiments have the fixed step size 25, why is that? 2. Why is the application hard to solve and how relevant it is to solve it at a certain accuracy? It is hard to me grasp the metric used regarding the clinical impact to the patient outcome. Overall, I think there're novelty in computational efficiency gain. But the experimental design and execution is rather weak.

Reviewer 3



Overall I think this is very nice work. The idea is clear and interesting. The mathematics are well formulated. The only drawback here is it seems the comparison experiments are somehow insufficient and therefore readers may not know the performance comparison with other more recent methods such as [4,5]. Also, I think the authors should release the code as an additional contribution. But overall I think the idea of step-wise alignment is interesting enough for this paper to be considered as a good submission.

[Author Response · NeurIPS 2019]

We would like to thank all reviewers for their time and effort writing these valuable reviews. The evaluation of different image modalities proposed by Reviewer 1 is an interesting idea, we will consider for future work.

Reviewer 3 mentioned that a performance measure with other recent methods would be beneficial. Most of the current methods in literature work on the registration of 3D image data. We will do further performance evaluation when we extend our method for the 3D case. The code for this paper will be released with the camera-ready version.

In the following, we focus on the questions given by Reviewer 2.

**1: The network contains fewer parameter than the B-spline method?**
The presented network does not contain fewer parameters compared to the classical B-spline method for optimization. However, the number of parameters to describe the final displacement field is smaller than with the B-spline registration, i.e. our method creates a compact representation of the displacement.

**2: Why sequence based image registration?**
The overall goal of the sequence-based registration approach is that the number of steps that are needed for the registration are not fixed from the beginning. The network should continue the registration until both images are correctly registered. In order to show that the idea of sequence-based registration is able to register two images in the first place, we use a fixed number of steps. For future work, we will investigate how we can integrate the stopping of the registration into the learning process.

**3: Why a Gaussian function as local transformation?**
We use a Gaussian function as local transformation as it provides an elegant method to modify the shape of the local transformation, i.e anisotropic local deformations. Furthermore, it is straightforward to extend for the 3D case. However, we are not limited to Gaussian functions.

**4: Why is the parameter network not explicitly made aware of the result of the position network?**
We decided to use a hierarchic approach for the estimation of the position on different spatial resolution in order to get an accurate estimation for the position. This idea is inspired by the hierarchic approaches used in classical image registration methods in order to avoid local minima during the optimization. However, for the parameter estimation, we think it is useful to place the parameter network at the end because with this we can take the global registration status into account. It might also have an additional regularization effect to prevent the parameter network from creating local extreme values, compared to the current registration status. It would be an interesting idea to add the parameter estimation for the different spatial resolutions too.

**5: Why is the registration of chest MRI images important and hard to solve?**
As mentioned in Section 4 the MR image data we used for the experiments are acquired with a special MR-sequence that is able to detect physiological changes inside the lung. For the analysis of the image data, a registration of corresponding anatomical structures is needed. Based on the registered image data it is possible to detect impaired regions of the lung. Using MRI allows continuous monitoring of the disease state which is critical for the treatment of chronic lung disease like cystic fibrosis, especially for pediatric patients.
The registration of chest images is a major topic in the field of medical image registration, because of the discontinuous displacements at the border of the lung and the thoracic cavity. Here, the major problem is that the thoracic cavity has a different motion direction compared to the lung. On the other hand, the displacement should be smooth inside the lung. Parametric transformation models like B-spline create smooth displacements but have problems at sliding organ boundaries. Non-parametric transformation models are able to create sharp discontinuities but have problems with the smoothness inside the lung. With our method, we can provide smooth displacements inside the lung and we are able to adapt the shape and the position of the local transformations at the lung border to increase the registration accuracy in this regions.

**6: Registration of longitudinal image data and other image modalities like MR/CT.**
As mentioned in question 5 the registration of thoracic images is challenging. Longitudinal registration or the registration of different modalities for 2D images is difficult as the slice positions may not be the same for two acquisitions and therefore different structures are acquired. Both experiments are very interesting, but we will leave this for future work together with a 3D version of the presented method.

**7: How to obtain a corresponding point given the current model?**
A point to point transformation is not directly possible with this model as it needs two images as input. After the registration is performed by the network a point in the moving image can be transformed with the final displacement. The final displacement is the sum of all local displacements.

**8: Detailed explanation of the B-spline method.**
The B-spline method belongs to the class of parametric registration methods. Here, the transformation of interest between the fixed and the moving image is approximated by a number of B-spline kernel functions located on a fixed regular grid over the image domain (Figure 2b). The fixed grid is used because the transformation of interest is not known in advance. The weights of the B-spline kernel functions are optimized using gradient based optimization methods for each image pair. A more detailed description is given in [22].

[Meta-Review · NeurIPS 2019]

The paper seems to contribute in a significant way in proposing an alternative RNN-based approach for deformable image registration. Although the experimental setting is not extremely strong, the proposed approach seems to give significant computational advantages. Rebuttal clarified most of the reviewers concerns.